# Foley Catheter as a Tourniquet for Peripartum Hemorrhage Prevention in Patients with Placenta Accreta Spectrum—A Two Case Report and a Review of the Literature

**DOI:** 10.3390/medicina59040641

**Published:** 2023-03-23

**Authors:** Jakub Staniczek, Maisa Manasar-Dyrbuś, Kaja Skowronek, Ewa Winkowska, Rafał Stojko

**Affiliations:** Department of Gynecology, Obstetrics and Gynecological Oncology, Medical University of Silesia, 40-055 Katowice, Poland

**Keywords:** placenta accreta spectrum, placenta, postpartum hemorrhage, Foley catheter

## Abstract

One of the most perilous complications in obstetrics, often leading to severe bleeding and sometimes a need for urgent hysterectomy, is placenta accreta spectrum, which significantly increases the risk of peri-partum complications, even including the risk of death for the mother and the child. Dealing with excessive bleeding in this situation is paramount. We have found a Foley catheter tourniquet to be useful as a temporary tourniquet to control placental and uterine hemorrhage. We have used this method and find it very useful. In this publication, we describe the last two cases of the use of the Foley catheter as a tourniquet for peri-partum hemorrhage prevention, and we will present a review of the literature in this field.

## 1. Introduction

One of the most perilous complications in obstetrics, often leading to severe bleeding and sometimes a need for urgent hysterectomy, is placenta accreta spectrum, which significantly increases the risk of peripartum complications, even including the risk of death for the mother and the child. As the bleeding from the uterus, which is extensively supplied with blood, requires almost immediate cessation, various methods of rapid ligation of the vessels supplying this organ exist. The methods available in the literature are often not reproducible, especially in the case of anatomical variability of the pelvic organs, and do not provide sufficient effectiveness or may require additional resources and costs. Therefore, there is a need for an alternative that is readily available, cheap, and usable even by inexperienced medical personnel.

According to the authors, the use of a Foley catheter as a tourniquet in order to limit hemorrhage during cesarean section due to PAS may be an alternative. Few publications describe this method. The authors managed to find six publications on this topic. The list of publications describing the use of a tourniquet-foley catheter for placenta accreta syndrome, and the size of the study group are presented in Table 1.

The publications describe the possibility of using one or two Foleys as a tourniquet; however, there is no consensus on the superiority of one method over another. In our publication, we present two cases of using the method with one Foley catheter and a review of the literature in this field.

## 2. The Method

The method used by the authors is an extension of the method proposed by Ikeda T et al. [1] It is worth emphasizing that at the stage of planning the operation or in an emergency, the tourniquet method can be combined with other hemostatic methods, such as embolization of the iliac vessels, hemostatic sutures, or the use of diathermy. When performing the placenta accreta syndrome surgery, a midline incision avoiding the umbilicus is the method of choice. In the case of a primary Pfannenstiel incision, the incision should be widened cephald in the midline. After opening the abdominal wall and visualizing the uterus, in the case of PAS, the incision of the uterus should be above the intrauterine margins of the placenta to minimize bleeding. It is good practice to perform an ultrasound examination before the incision to locate the opening of the uterus. According to our assumptions, after delivery of the fetus, we extract the uterus from the abdomen by gently grasping the fundus of the uterus and pulling upward and forward. Ensure that the uterine appendages are released on both sides by moving the uterus to the right and left. (Figure 1 and Figure 2). In another stage, an assistant uses a sterile Foley catheter (Ch 16/18 French) to slide it down (caudal) to the lowest point and fix it “en bloc” around the cervix at the level of the uterosacral ligaments, about 3–4 cm below the level of the incision (Figure 3 and Figure 4). Then tighten it and fix it with Kocher forceps so that the tourniquet does not loosen (Figure 5 and Figure 6). Hemostasis achieved with the tourniquet technique allows the operator time to evaluate the possibility of preserving the uterus. In order to evaluate the active bleeding, the tourniquet can be temporarily loosened. This maneuver also gives time to prepare a blood transfusion or call for support. The tourniquet method may also be used as a primary management strategy for PAS or as a follow-up management strategy after placental removal and subsequent hemorrhage.

## 3. Case Reports

### 3.1. Patient 1

We present two cases of patients with PAS who underwent emergency cesarean sections in our clinic. In the first case, a 29-year-old woman was admitted to the department with an ultrasonographic diagnosis of placenta previa and PAS FIGO Grade 2. The diagnosis was confirmed by magnetic resonance imaging (MRI) in the 27th week of pregnancy. MRI has shown “placenta previa passing partly through the front wall of the uterus and segmental thinning and blurred outline of the anterior wall of the uterus in the scar after cesarean section, which corresponds to the ingrowth of the placenta into the scar”. An elective cesarean section and hysterectomy were planned for the 36th week of pregnancy, according to the standards of the Polish Society of Gynecologists and Obstetricians. The patient did not suffer from any chronic diseases; she underwent laparoscopic cholecystectomy in 2015 and one previous on-term cesarean section because of a lack of progress in labor. The patient had antepartum bleeding in the 24th week of pregnancy, which was a risk factor for an emergency cesarean section [7]. In the 31st week of pregnancy, the patient presented with a massive uterine hemorrhage. The patient was diagnosed with a premature detachment of the placenta and qualified for an emergency cesarean section. After general anesthesia, a longitudinal incision avoiding the navel was made, and the uterus has been exteriorized. Upon visualization of the uterus, it was noted that the placenta had grown through the previous cesarean section scar. A transverse incision of the uterus was performed above the placenta. A female fetus weighing 1666 g and measuring 43 cm in length was extracted from the cephalic position (8/8/8/9 points in the Apgar score and pH 7.345). The cord was clamped. Before attempting to remove the uterus, we used the tourniquet method with a sterile Foley catheter (18 French). The Foley catheter was slid down (caudally) to the lowest point, fixed by the assistant, and then tightened and fixed with Kocher forceps at the level of the uterosacral ligament to stop the bleeding (Figure 7 and Figure 8). Further stages of the operation proceeded in a typical manner for a hysterectomy. The operation lasted 78 min. The blood loss measured immediately after surgery was 4.3 g/dL (12.3 g/dL to 8.2 g/dL). During surgery, the patient required three units of red cell concentrate transfusion. The estimated blood loss was 3000 mL, and no complications occurred during the postpartum period. The patient and her child were discharged home on the fourth postoperative day. A histopathological examination confirmed PAS and confirmed that the placental plate exceeded the entire thickness of the uterus.

### 3.2. Patient 2

The second case describes a 32-year-old patient in her second gestation who did not suffer from any chronic diseases and had not undergone any operations. She had one vaginal birth with postpartum uterine curettage due to a retained placenta. The patient had a previous bleeding episode in the 26th week of pregnancy and was diagnosed with central placenta previa with a “placental bulge sign”, which suggested the possibility of growing into the front uterine wall. The patient received prenatal steroid therapy (two doses of betamethasone), and an MRI appointment was scheduled at the 30th week of gestation. In the 28th week of pregnancy, the patient presented to the hospital with vaginal bleeding. During a short stay in the hospital, increased bleeding and regular contractions were observed. Gynecological examination revealed a cervical dilatation of 3–4 cm with a visible placenta and bleeding at the cervical os. A fetal doppler showed an umbilical artery pulsatility index of 1.54 (>95%), a ductus venosus pulsatility index of 0.85 (>95%), and a cardiotocographic short-term variation of 2.1 msec. Neuroprotection was started, and the patient qualified for an urgent cesarean section. After general anesthesia, a Pfannenstiel incision was made, and the uterus has been exteriorized. Upon visualization of the uterus, a transverse incision of the uterus was performed above the placenta. A female fetus weighing 1100 g and measuring 42 cm in length was extracted from the breech position (1/1 point in the Apgar score and pH 6.950). The cord was clamped. Before attempting to remove the placenta, the tourniquet method with a sterile Foley catheter (18 French) was performed. The Foley catheter was slid down (caudally) to the lowest point, fixed by the assistant, and then tightened and fixed with Kocher forceps at the level of the uterosacral ligament to mitigate the bleeding (Figure 9 and Figure 10). With the Foley catheter clamped, the placenta was removed, and small placental increta sites were excised. The Foley catheter tourniquet was slowly released to ensure there was no excessive bleeding. The remaining stages of the cesarean section were uneventful. The surgery lasted 45 min. The estimated blood loss during the operation was about 500 mL. Immediately after surgery, it was 0.4 g/dL (10.4 g/dL before admission, 8.7 g/dL just before the cesarean section, and 8.3 d/dL after surgery). The patient did not require a red cell concentrate transfusion, as the estimated blood loss was 500 mL. No complications occurred during the postpartum period. The patient was discharged home on the second postoperative day.

## 4. Discussion

The placenta accreta spectrum includes placenta accreta, increta, and percreta. PAS is a complex obstetric complication that leads to high morbidity and mortality in both the mother and the neonate. The incidence of PAS is constantly increasing due to various causes, including the increasing numbers of cesarean sections, advanced maternal age, and an increasing use of assisted reproductive techniques, which have all been identified as risk factors for incorrect placental implantation [8,9]. The development of PAS involves the defective decidualization of the uterine lining, resulting in an inadequate formation of the decidua basalis, the layer of the endometrium that separates the placenta from the myometrium. This defective decidualization leads to invasion of the trophoblast into the myometrium, eventually leading to PAS, which in turn results in failure of the placenta to detach from the uterine wall during delivery and increases the risk of significant bleeding. In the prior years, several surgical techniques were developed to reduce the risk of profuse peripartum bleeding, including uterine arterial ligation, occlusion of the uterine arteries, and various methods of coagulation; however, all have their limitations, being either irreversible (in the case of definite arterial ligation) or insufficiently effective (in cases of excessive bleeding).

According to the standards of the Polish Society of Gynecologists and Obstetricians, it is recommended that patients diagnosed with PAS have an elective cesarean section between the 34th and 37th week of pregnancy, which is estimated to achieve the most optimal fetal growth and reduce the risk of an emergency cesarean section [10]. It has been demonstrated that in the overall population of pregnant patients, the cesarean section performed in an elective setting is associated with a reduced risk of bleeding when compared with emergency procedures [11]. Thus, in the setting of PAS, the delivery, which is already associated with an increased risk of periprocedural complications, should be most appropriately scheduled before the risk of an emergency procedure increases after the 37th week.

Our case reports and further studies related to the use of a Foley catheter as a tourniquet aim to define a new method for reducing postpartum bleeding associated with PAS. Different types of PAS carry different risks of intraoperative hemorrhage, with a tendency for a higher risk of significant bleeding in the placenta percreta. Although, due to difficulties with establishing definite diagnoses and differentiating between specific subtypes of PAS, all manifestations of PAS should be considered to be at high risk of peripartum bleeding. Placental blood supply is an important factor that influences the frequency of intraoperative bleeding and hysterectomy [12]. In the study by Meng et al., two tourniquets were used during cesarean sections to stop bleeding in patients with PAS, thus allowing more time to dissect the placenta and maintain hemostasis. The mean blood loss during the procedure was 1286 ± 175 mL, and the uterus was preserved in 18 of the 23 (78.3%) patients who received this hemostatic technique [4]. Ikeda T et al. used a tourniquet in two out of four patients with incorrect placental implantation, achieving significantly less blood loss compared to the other two patients—of whom one had traditional hemostatic sutures and the other required a hysterectomy to stop intraoperative bleeding. In the same publication, Ikesa T. et al. suggested the possibility of pinching both infundibulopelvic ligaments with Satinsky hemostatic forceps and adding a rubber tube to perforate the avascular area of the broad ligament under the round ligaments [1]. In our experience, this is more difficult to perform, especially in the case of multiple adhesions or placenta percreta that extends to extrauterine organs such as the bladder, broad ligament, or bowels. In turn, Envian et al. used a Foley catheter as a tourniquet in a 35-year-old patient after three cesarean sections with PAS in a subsequent pregnancy. Owing to this maneuver, it was possible to reduce bleeding and preserve the uterus without complications in the postoperative period [2]. The studies described above showed good results with compression tourniquets used to stop bleeding in PAS. The use of tourniquets may reduce the blood supply to the placenta during surgery, thereby reducing intraoperative bleeding [3]. Thus, it can be concluded that this technique is an effective and safe intervention to control postpartum hemorrhage during cesarean section due to PAS as a first step to avoid massive bleeding. Thus, early cessation of excessive bleeding increases the comfort of the operators, enhances the visibility in the operated field, and thus might reduce the risk of damage to the neighboring organs.

It is worth noting that some publications describe techniques with two tourniquets [4]. From our experience, we see the optimal effects of applying only one Foley catheter as a tourniquet, additionally fixed with Kocher forceps at the level of the uterosacral ligament, which prevents the catheter from untying. Furthermore, a single band is more cost-effective and easier to put on, thus giving the operators the opportunity to plan subsequent procedures during surgery, such as blood transfusion or assessment of the possibility of uterine preservation [5]. Easy access to Foley catheters, which are widely used in any labor room, and their low cost make it a method that can be used in any obstetric ward, and with proper planning, it can be combined with more advanced methods of stopping obstetric hemorrhage. Finally, our study has some limitations that one should be aware of. First, the case report describes two patients, so drawing conclusions must be performed with caution. Second, in the study, no comparison with other, widely used methods of bleeding mitigation, such as typical arterial ligation, was performed. Moreover, both cases describe an urgent cesarean section; therefore, a benefit of the method during an elective procedure requires verification. Finally, the anatomical variability of the arterial vessels supplying the uterus might in some situations prohibit the use of a single Foley catheter as a tourniquet, especially in cases when the uterine artery divides in its most proximal part, and thus the use of multiple ligation equipment might be necessary. The list of publications describing the use of a tourniquet-foley catheter for placenta accreta syndrome, and the size of the study group are presented in Table 1.

## 5. Conclusions

The use of a Foley catheter as a tourniquet is a widely used method in our department. There are few publications on this subject in the literature. We believe that the topic should be further explored. The technique seems to be an effective method of prevention to reduce bleeding in a patient with incorrect placental implantation. It is also useful during perinatal hysterectomies. It gives surgical team members time to prepare the appropriate procedure. It reduces blood loss during the procedure, which improves the operating environment and the patient’s condition after surgery. We expect that it will also generate lower costs related to the patient’s hospitalization—a shorter hospital stay and less frequent red blood cell and plasma transfusions. It is worth emphasizing that the method we propose is simple and cheap to perform. Foley catheters are available in every maternity ward.

## Figures and Tables

**Figure 1 medicina-59-00641-f001:**
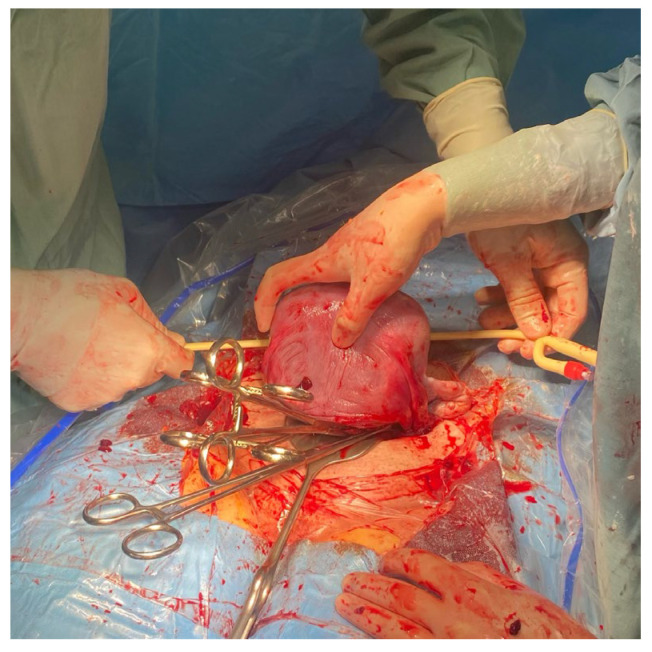
The assistant extracts the uterus from the abdomen by gently grasping the fundus of the uterus and pulling it upward and forward.

**Figure 2 medicina-59-00641-f002:**
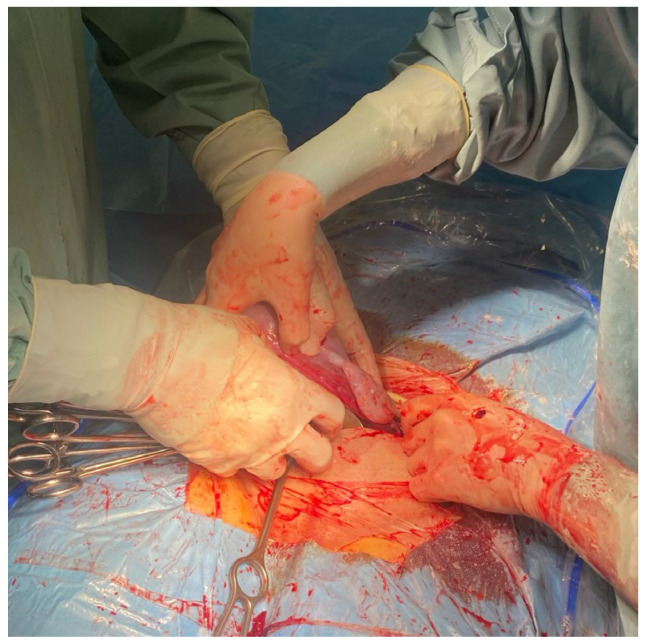
The assistant ensures that the uterine appendages are released on both sides by moving the uterus to the right and left.

**Figure 3 medicina-59-00641-f003:**
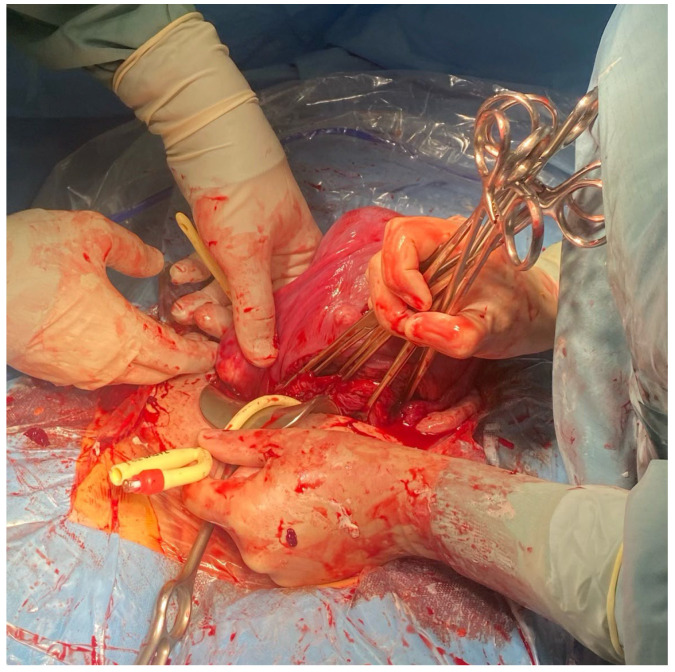
The assistant uses a sterile Foley catheter (Ch 16/18 French) to slide it down (caudally) to the lowest point, at the level of uterosacral ligaments.

**Figure 4 medicina-59-00641-f004:**
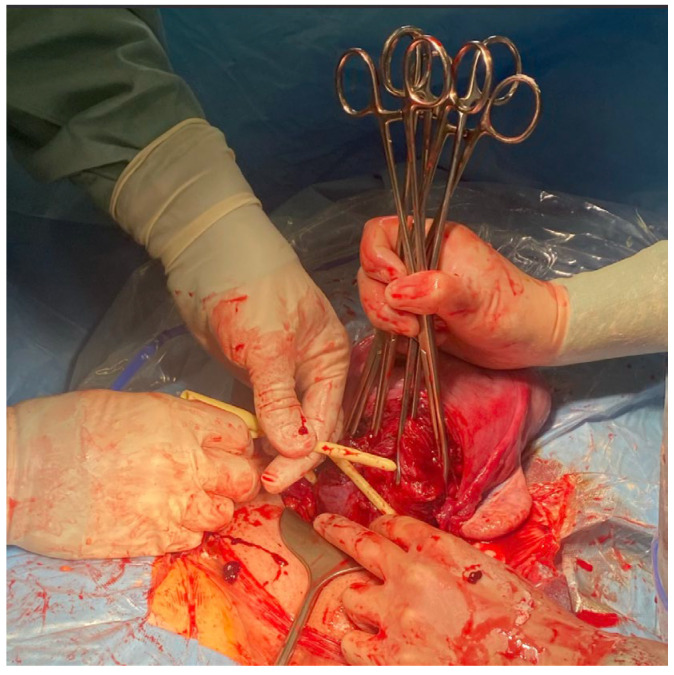
The assistant fixes a sterile Foley cathether “en bloc” around the cervix at the level of the uterosacral ligaments, about 3–4 cm below the level of the incision.

**Figure 5 medicina-59-00641-f005:**
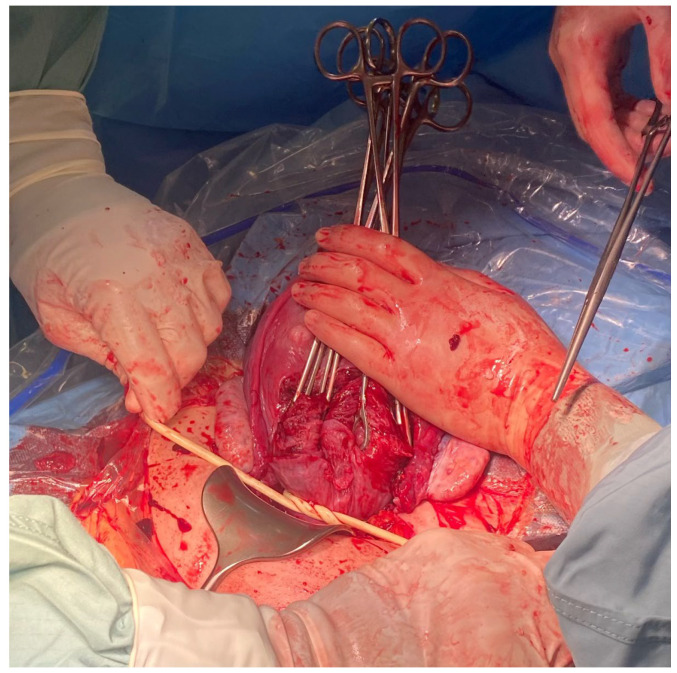
The assistant tightens the Foley catheter.

**Figure 6 medicina-59-00641-f006:**
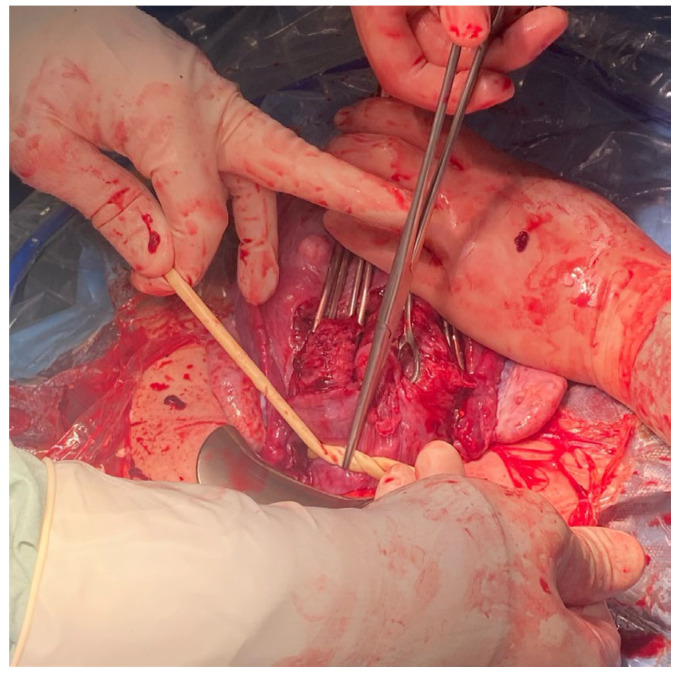
The assistant fixes the Foley catheter with Kocher forceps.

**Figure 7 medicina-59-00641-f007:**
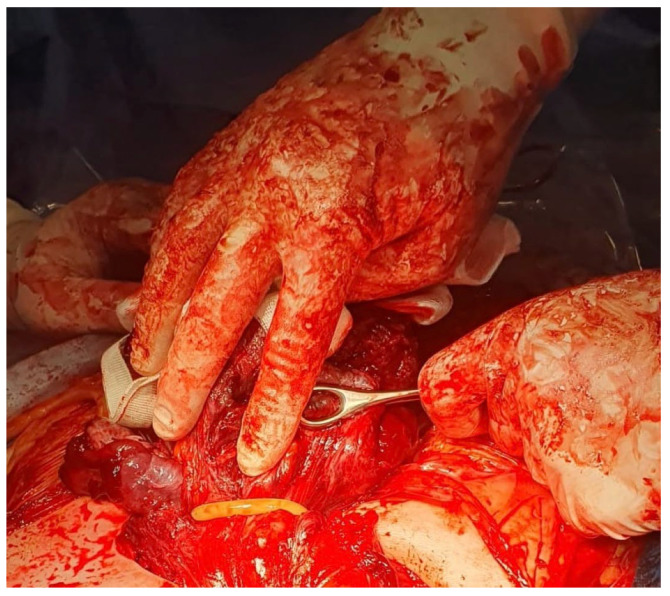
Placement of a Foley catheter as a tourniquet to prevent hemorrhage.

**Figure 8 medicina-59-00641-f008:**
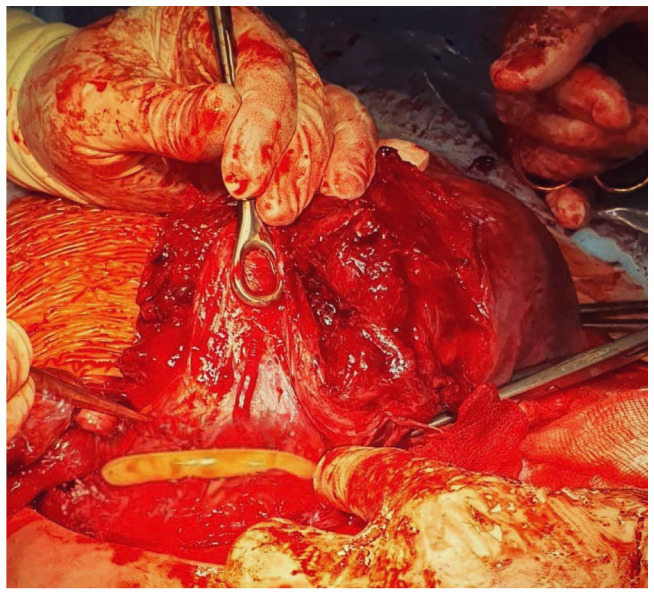
Placement of a Foley catheter as a tourniquet to prevent hemorrhage.

**Figure 9 medicina-59-00641-f009:**
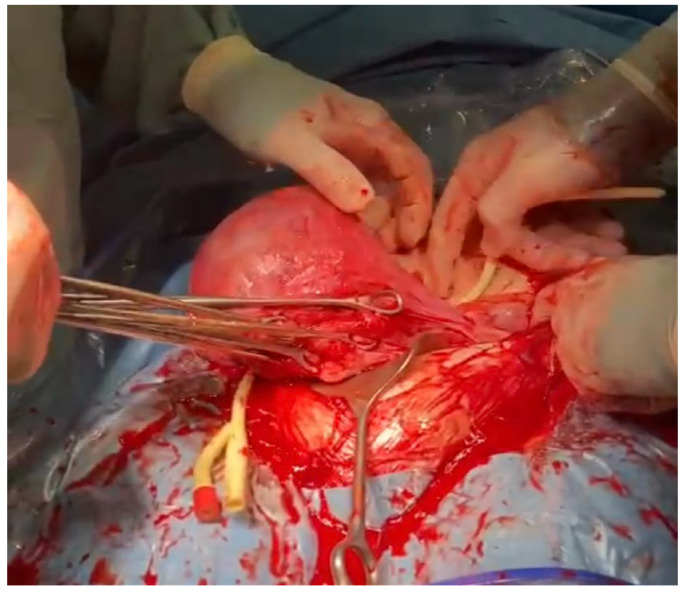
Placement of a Foley catheter as a tourniquet to prevent hemorrhage.

**Figure 10 medicina-59-00641-f010:**
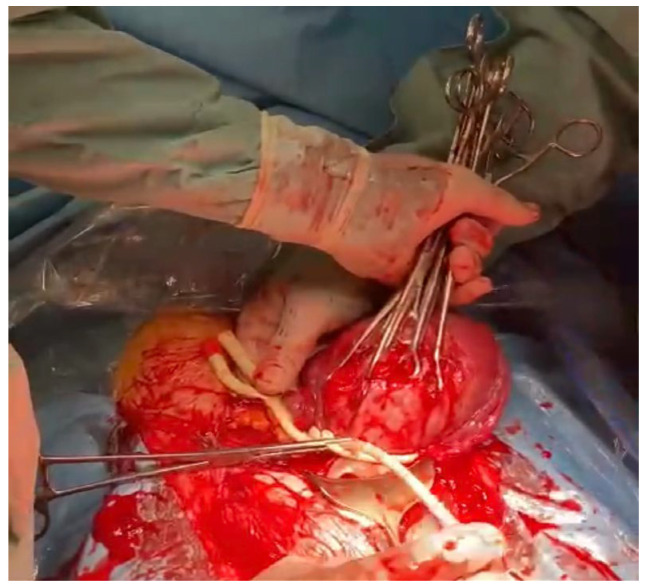
Placement of a Foley catheter as a tourniquet to prevent hemorrhage.

**Table 1 medicina-59-00641-t001:** List of publications describing tourniquet-foley catheters for placenta accreta syndrome.

Authors	Study Group	Date of Publishing
Ikeda T et al. [1]	4	2005
Envain F et al. [2]	1	2020
Huang J et al. [3]	559	2021
Meng JL et al. [4]	20	2017
Abdelaziz AT et al. [5]	21	2012
Altal OF et al. [6]	11	2022

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
