# Peer review of "Foley Catheter as a Tourniquet for Peripartum Hemorrhage Prevention in Patients with Placenta Accreta Spectrum—A Two Case Report and a Review of the Literature"

_medicina, 2023, doi:10.3390/medicina59040641_

Round 1

Reviewer 1 Report

Dear authors

I have assessed your case report of 2 cases of PAS exploring the option of Foleys catheter as a tourniquet to reduce intraoperative blood loss during cesarean/hysterectomy.

Although it appears to be an easy technique, I have few observations.

1. Authors themselves have cited studies where previous authors have described similar technique with 2 catheters, here authors have used only 1. The argument against your technique could be that it's possible that 1 catheter loosens during the procedure and thus 2 provides better homeostasis! Authors need to defend their technique against 2 catheters.

2. Secondly, since it's a description of just 2 cases, not much scientific conclusion can be drawn, especially if authors have 'over a dozen' such cases. Why not report a 'case-series'?

3. Lastly, without a comparator arm, it cannot be concluded that this techniques actually reduces blood loss, and to what extent! Yes it's logical that any compression over uterine arteries will reduce blood loss, but the extent of it is only speculative

Author Response

Dear Madam/Sir,

Thank you very much for your evaluation. I will answer the questions in detail:

1. In our method, after clamping the Foley catheter, we clamp it with Koher forceps. There is no way for such a tightening to release on its own.

2. We are at the stage of evaluating our cases to standardize the method and present a comparison in the future.

3. In the literature there are comparisons, for example, to clamp the uterine arteries by pressing with the hand or with atramatic forceps. The compression method as an emergency seems to work. Comparison will come as we get more cases.

Thank you very much for your comments. They will be in the manuscript.

Yours faithfully,

Reviewer 2 Report

Thank you very much for giving me the opportunity to review this case report. My specific comments regarding the manuscript are included below;

- There are lots of grammatical errors that should need to be corrected (i.e. which significantly increases, the preferred methods, provide complete efficacy, a woman aged, the ultrasonographic diagnosis of, missing commas, etc.). The language of the manuscript should be improved.

- All abbreviations should have been provided in full on the first mention and this applies to the title, running (short) title, abstract, impact statement, main text, and each table/figure independently as they will be read independently. Please use the abbreviations correctly and effectively. So, all the abbreviations should be checked (i.e. MRI, Uma PI, DV, etc.).

- Recently, Oglak et al. concluded that three factors-antepartum bleeding episode during pregnancy, first antepartum bleeding episode ≤28 weeks of gestation, and lower preoperative hemoglobin level-might be useful in predicting emergency cesarean delivery in pregnancies complicated with placenta previa (Oğlak SC, Ölmez F, Tunç Ş. Evaluation of Antepartum Factors for Predicting the Risk of Emergency Cesarean Delivery in Pregnancies Complicated With Placenta Previa. Ochsner J. 2022;22(2):146-153. doi: 10.31486/toj.21.0138). Are any of these cases admitted to the hospital with any of these complaints or findings? Please discuss it in the discussion section.

- How did you schedule the delivery timing of these cases? Please discuss it in the discussion section.

Author Response

Dear Madam/Dear Sir,

Thank you very much for your opinion. I will answer the questions exactly:

1. The manuscript has already undergone another translation, I hope that the errors have been corrected.

2. A list of abbreviations has been added to the manuscript.

3. Thank you very much for this comment, the discussion will be supplemented with this publication and supplemented with missing information.

4. Termination of pregnancy with PAS is regulated in Poland by the standard of the Polish Society of Gynecologists and Obstetricians. It talks about the end of such a pregnancy between 34-37 weeks. I will add it in discussion.

Yours faithfully

Round 2

Reviewer 2 Report

Thank you for the revisions

Author Response

Dear Madam/Sir,

I add manuscript. We I made changes.

Regards,

Jakub Staniczek
